# Assessing the effect of compounds from plantar foot sweat, nesting material, and urine on social behavior in male mice, *Mus musculus*

**Amanda J. Barabas**[1], **Helena A. Soini**[2], **Milos V. Novotny**[2], **Jeffrey R. Lucas**[3], **Marisa A. Erasmus**[1]*, **Heng-Wei Cheng**[4], **Rupert Palme**[5], **Brianna N. Gaskill**[1]

**1** Department of Animal Science, Purdue University, West Lafayette, Indiana, United States of America,
**2** Department of Chemistry, Indiana University, Bloomington, Indiana, United States of America,
**3** Department of Biological Science, Purdue University, West Lafayette, Indiana, United States of America,
**4** USDA-ARS, Livestock Behavior Research Unit, Purdue University, West Lafayette, Indiana, United States of America, **5** Unit of Physiology, Pathophysiology, and Experimental Endocrinology, University of Veterinary Medicine, Vienna, Austria

* merasmus@purdue.edu

**Data Availability Statement:** All relevant data are within the article and its Supporting Information files.

## Abstract

Home cage aggression causes poor welfare in male laboratory mice and reduces data quality. One of the few proven strategies to reduce aggression involves preserving used nesting material at cage change. Volatile organic compounds from the nesting material and several body fluids not only correlate with less home cage aggression, but with more affiliative allo-grooming behavior. To date, these compounds have not been tested for a direct influence on male mouse social behavior. This study aimed to determine if 4 previously identified volatile compounds impact home cage interactions. A factorial design was used with cages equally split between C57BL/6N and SJL male mice (N = 40). Treatments were randomly assigned across cages and administered by spraying one compound solution on each cage's nesting material. Treatments were refreshed after day 3 and during cage change on day 7. Home cage social behavior was observed throughout the study week and immediately after cage change. Several hours after cage change, feces were collected from individual mice to measure corticosterone metabolites as an index of social stress. Wound severity was also assessed after euthanasia. Measures were analyzed with mixed models. Compound treatments did not impact most study measures. For behavior, SJL mice performed more aggression and submission, and C57BL/6N mice performed more allo-grooming. Wound severity was highest in the posterior region of both strains, and the middle back region of C57BL/6N mice. Posterior wounding also increased with more observed aggression. Corticosterone metabolites were higher in C57BL/6N mice and in mice treated with 3,4-dimethyl-1,2-cyclopentanedione with more wounding. These data confirm previous strain patterns in social behavior and further validates wound assessment as a measure of escalated aggression. The lack of observed treatment effects could be due to limitations in the compound administration procedure and/or the previous correlation study, which is further discussed.

**Funding:** This work was funded by a grant awarded to B.N.G, M.A.E., J.R.L., H.W.C., H.A.S., and M.V.N. by the Purdue University Center for Animal Welfare Science. Also, GC-MS data was analyzed using the Aligent GC-QTOF-MS purchased by NSF grant #1726633, awarded to Jonathan A. Karty. The funders had no role in study design, data collection and analysis, decision to publish, or preparation of the manuscript.

**Competing interests:** The authors have declared that no competing interests exist.

# Introduction

Aggression among group housed male laboratory mice continues to challenge researchers despite its negative impacts on animal welfare and research data quality [1,2]. Although aggression is a complex social situation caused by a variety of factors [1,3,4], it is often suggested that odor signals could appease conflict since they are a natural form of communication for many mammalian species [5–7]. For mice specifically, aggression can be triggered by scent cue disruption [8]. For example, the routine cage cleaning that mice experience can often cause bouts of violent, escalated aggression that peak approximately 15–45 minutes afterward [9,10]. One of the few proven remedies for aggression related to cage change is transferring used nesting material into the new cage [11], and for decades it has been speculated that this mechanism is due to odor signals preserved in the material. Recently it has been confirmed that used nesting material does in fact contain a variety of proteins used by mice for identification purposes [12], so the practice of transferring used nesting material is supported by an ethologically relevant form of communication.

Specifically, it has been suggested that mice deposit pheromones in nesting material that appease aggression among familiar conspecifics. Pheromones are a subcategory of odor signals that must meet specific criteria for classification. For instance, an odor signal must produce reliable effects in a bioassay at physiologically relevant concentrations to be considered a pheromone [6,13]. In mice, the only known pheromones that impact same sex social behavior are those produced in urine that promote inter-male aggression [14–17]. In general, research on mammalian odor signals is dominated by urinary compounds that promote aggression [18]. However, preliminary work has shown that geranylacetone detected in used nesting material has a negative correlation with home cage aggression [19]. This compound has also been found in murine saliva and plantar sweat [19] and the ventral gland of hamsters, which is typically used for marking territory [20,21]. To the best of our knowledge, it has not been tested for a direct behavioral role in mice.

While minimizing home cage aggression would improve the welfare of laboratory mice, it is only the bare minimum that could be done for the animals' social environment. Promoting positive affect and pleasurable emotional states is one key component of good overall welfare [22]. Since mice are naturally a social species [23], their welfare would be greatly enhanced if socio-positive/affiliative behaviors could be promoted in captivity. However, it has also been suggested that affiliative behaviors can play a context dependent role in resource control, proving more beneficial in situations with abundant resources, such as in the laboratory [24]. Unfortunately, there is a lack of fundamental knowledge on how specific odors directly impact affiliative behaviors: in a scoping review focused on how odor signals impact terrestrial mammalian social behavior, less than 2% of reported behavioral measures were affiliative [18]. For mice, most work on captive social behavior focuses on aggression between unfamiliar males, leaving affiliation in the home cage overlooked. A key murine affiliative behavior is allo-grooming, which is often done to strengthen social bonds [25]. Preliminary work found that three volatile organic compounds (VOC) correlate with allo-grooming in group housed male mice: 3,4-dimethyl-1,2-cyclopentanedione, 3,5-diethyl-2-hydroxycyclopent-2-en-1-one, and 6-hydroxy-6-methyl-3-heptanone [19]. The two cyclopentanone compounds have never been tested for a direct animal behavior role and appear to be unique to murine plantar sweat glands [19]. Plantar sweat does not have a confirmed role in terms of social interactions, but it has been associated with territory marking and colony member recognition [25,26]. On the other hand, 6-hydroxy-6-methyl-3-heptanone is found in male mouse urine and is known to accelerate puberty in female mice [27]. However, it has never been tested for a role between male mice.

This study served as a follow up to previous work demonstrating a correlation between four VOCs and social behavior in group housed male mice [19]. All four VOCs show potential to be murine pheromones, but must undergo more stringent testing to be considered so [6,13]. Therefore, the goal of this study was to examine the direct role of geranylacetone, 3,4-dimethyl-1,2-cyclopentanedione, 3,5-diethyl-2-hydroxycyclopent-2-en-1-one, and 6-hydroxy-6-methyl-3-heptanone on murine social behavior. We hypothesized that all four compounds could act as murine pheromones and alter social behavior. We had two predictions: first, geranylacetone would reduce aggression in the home cage; second, 3,4-dimethyl-1,2-cyclopentanedione, 3,5-diethyl-2-hydroxycyclopent-2-en-1-one, and 6-hydroxy-6-methyl-3-heptanone would increase allo-grooming among familiar male mice. In addition to social behavior, subcutis wounding was examined as a secondary aggression measure and fecal corticosterone metabolites were assessed as an index of social stress.

## Methods

### Ethics statement

Animal procedures were approved by Purdue University's Institutional Animal Care and Use Committee (protocol # 1707001598). Humane endpoint criteria were established for cages displaying excessive aggression. Any mouse with wounding greater than $1cm^2$ would be immediately euthanized. Cages were monitored daily for wounding, signs of pain/distress, and general activity. Welfare checks occurred within two hours of the mice's active period to identify any wounding as quickly as possible. No cages met these criteria.

### Treatment preparation

Three of the four compounds were obtained from commercial vendors and were stored according to manufacturer recommendations: geranylacetone and 3,4-dimethyl-1,2-cyclopentadione (Sigma- Aldrich, St. Louis, MO); 6-hydroxy-6-methyl-3-heptanone (Chemspace, Monmouth Junction, NJ). 3,5-diethyl-2-hydroxycyclopent-2-en-1-one was synthesized at Indiana University (Bloomington, IN) using previously described methods [19] and was kept in a -80˚C freezer when not in use. Test solutions of each compound (i.e., one individual compound per solution) were formed based on natural concentrations that correlate with either lower levels of aggression or higher levels of affiliative behavior [19]. The maximum compound weight previously detected in a single sweat or urine sample was adjusted to represent five mice per cage and used to calculate the concentrations for this study. The final concentrations are reported in S1 Table. However, we acknowledge that it is unknown if levels of 3,4-dimethyl-1,2-cyclopentadione and 3,5-diethyl-2-hydroxycyclopent-2-en-1-one are natural since pilocarpine was previously used to stimulate sweat production and it is unknown how compound values were affected [19]. Stock solutions were made with ethanol (Thermo Fisher Scientific, Waltham, MA), and were further diluted to natural concentrations in a 3% polyethylene glycol (PEG; Sigma- Aldrich), acetone (Thermo Fisher Scientific) solution. All ethanol stocks were stored at -80˚C and acetone test solutions were stored in a refrigerator.

In order to determine how long treatments would be detectable in the cage, test solutions were administered to empty mouse cages containing chow, water, aspen bedding, and crinkle paper nesting material. Samples from the cages with the test solution were compared to samples from cages with a control solution (3% PEG in acetone only) to detect increased levels of the test compounds in the cage headspace. Test and control cages were sampled in adjacent, positive pressure rooms. First, 100μL of the solutions were applied to a square of clean medical gauze placed in a metal tea ball (Shuo, Novi, MI) that rested on top of the wire food hopper. Samples of the cage headspace were collected using Twister™ polydimethylsiloxane coated

stir bars (Gerstel USA, Linthicum, MD) on days 1, 3, 5, and 7 after treatment application. One stir bar was placed at each end of each cage in a metal tea ball and suspended from the wire food hopper for eight hours on each collection day (S1 Fig). Stir bars were analyzed using gas chromatography- mass spectrometry (see below, "Gas chromatography- mass spectrometry").

Using natural concentrations, the test compounds were not elevated in the cage headspace compared to the control. Therefore, the compound concentrations were increased by 5x (S1 Table), and the procedure was repeated. The 5x concentration was sufficient to see increased levels of 6-hydroxy-6-methyl-3-heptanone in the headspace on collection days 1 and 3. The other three test compounds were not detectable in the headspace on any collection day. However, geranylacetone is a liquid at room temperature while 3,4-dimethyl-1,2-cyclopentadione and 3,5-diethyl-2-hydroxycyclopent-2-en-1-one are solids at room temperature, so the compounds likely retained these physical forms on the medical gauze instead of diffusing into the headspace.

Consequently, the administration route was changed, and the solutions were applied to the nesting material, so the mice could be in direct contact with the compounds (see below, "Treatment administration). Extractions from the treated nesting material were not tested as the processing chemicals in the material would have likely masked the compounds of focus. However, the treated nesting material's headspace was analyzed (see below, "Gas chromatography- mass spectrometry") and increased levels of 6-hydroxy-6-methyl-3-heptanone were detected on days 1 and 3 after treatment. For application consistency, all the test solutions were given to the mice at 5x natural concentrations and refreshed after 3 days.

## Gas chromatography- mass spectrometry

All sample processing and analysis took place at the Indiana University Mass Spectrometry Facility (Bloomington, IN). Samples of nesting material were stored in Ziploc bags and refrigerated at 4°C. Samples were analyzed on the same day they were received. The procedure was started within an hour of receipt from Purdue University. Approximately 0.58 g of each nesting material sample was placed into a clean 20 mL headspace vial. A previously conditioned and cleaned Twister™ PDMS coated stir bar (10 x 0.5 mm, Gerstel USA, Linthicum, MD) suspended in a glass headspace vial adapter (Gerstel USA) and the vial was sealed with a new screw cap containing a PTFE-silicone septum (Restek Corp, Bellafonte, PA). The vials were left at room temperature for 1 hour.

All Twister™ stir bars (both those that were suspended in the test cages and those that were in vials with the nesting material) were placed in standard 7" desorption tubes and desorbed using Gerstel TDSA2 autosampler feeding a TDU 3 thermal desorption unit (Gerstel). Each Twister™ was flushed with 52 mL/min of He and was heated at 60°C/min to 270°C and held at 270°C for five minutes. The gas stream was directed into a Gerstel CIS-4 programmable temperature vaporizer inlet held at -80°C throughout the desorption process. The condensed sample molecules were introduced into an Agilent 7890B gas chromatograph (GC) by heating the cooled injection system (CIS4) at 12°C/sec to 270°C and holding at 270°C for five minutes. The GC was set to solvent vent mode, and 23.573 psi was held in the inlet for 1.2 minutes. The GC column was a 30 m long, 250 μm inner diameter Agilent DB-5ms column with a 0.25 μm thick stationary phase. The oven was held at 40°C for one minute and then ramped at 2°C/min to 180°C followed by a ramp at 10°C/min to 270°C and held at that temperature for six minutes. The total cycle time was 86 minutes. An Agilent G7250B quadrupole-time-of-flight mass spectrometer served as the detector using a 70 eV electron ionization source. Mass spectra were recorded from m/z 41–400 at 5 scans/sec. Individual extracted ion chromatograms for each of the compounds were extracted using version 10.0 of Agilent Qualitative Analysis for GC-TOF.

## Animals and housing

A factorial design was used based on the five solutions (four VOC test solutions and 3% PEG, acetone control) and two mouse strains. One hundred male mice of each of the SJL/JOrlIcoCrl (SJL- Wilmington, MA) and C57BL/6NCrl (B6- Raleigh, NC and Kingston, NY) strains from Charles River were used (200 mice total). These strains were chosen based on correlation data from previous work [19]. Mice arrived at 8 weeks of age and were housed in open top cages (11.5" x 7.25" x 4.25"; Ancare, Bellmore, NY) in groups of five for a one-week study period (N = 40 cages). This sample size was determined *a priori* using Mead's resource equation [28]. All cages contained aspen bedding (Envigo, Indianapolis, IN), 8g of crinkle paper nesting material (Enviro-dri, Fibercore, Cleveland, Ohio), and *ad libitum* food (Envigo, Teklad 2018) and water. A 12:12 light cycle was used throughout the study (lights on at 6:00). All mice were ear punched for identification and randomly allocated into cages upon arrival using a sequence from RANDOM.org. All mice were weighed at arrival and the end of the study. On average, mice were 21.70 ± 1.86g at arrival and 22.00 ± 2.26g at sacrifice. For further details about euthanasia, please refer to the "Wounding" subsection of the methods.

Odor treatments cannot be administered in the same room due to cross contamination risk. Therefore, two rooms, each in a different building, were used in an incomplete block design: each solution was tested in each room, at different times, but the same solution was never tested concurrently in both rooms. Both facilities were located on Purdue University's West Lafayette, IN campus. Rooms in different facilities were intentionally chosen to examine if the treatments could overcome potential behavioral variation across facilities [29]. Major parameter differences between the facilities are outlined in Table 1. Since only two rooms were used at one time, mice arrived in five batches of forty, equally split between strains (40 mice/5 mice per cage; n = 8 cages per batch; 4 cages per room).

## Treatment administration

Treatment order for each room was randomly assigned using a RANDOM.org list generator (S2 Table). Each treatment solution contained one compound. Even though these compounds are naturally mixed in the environment, compound combination treatments could not be tested due to available resources. Wash out periods between treatments lasted at least one week. Treatment solutions were administered using an opaque 5mL glass spray bottle (Your Oil Tools, Hooksett, NH). Approximately 120μL of each treatment were applied to the 8g of nesting material before the mice were allocated to their cages. Based on personal consultation

**Table 1. Outline of parameter differences between housing rooms in different facilities.**

|  | Facility A | Facility B |
|---|---|---|
| **Temperature high interquartile range** | 22.22–22.78˚C | 23.33–23.89˚C |
| **Temperature low interquartile range** | 21.11–21.67˚C | 20.56–21.11˚C |
| **Humidity high interquartile range** | 43.5–50% | 51–57% |
| **Humidity low interquartile range** | 30–40% | 30–43.5% |
| **Air changes per hour** | 9.5 | 20.1 |
| **Water** | Reverse osmosis | Tap water |
| **Species on the same floor** | Mice and pigs | Mice and rats |
| **Care staff sex** | Female only | Male and female |

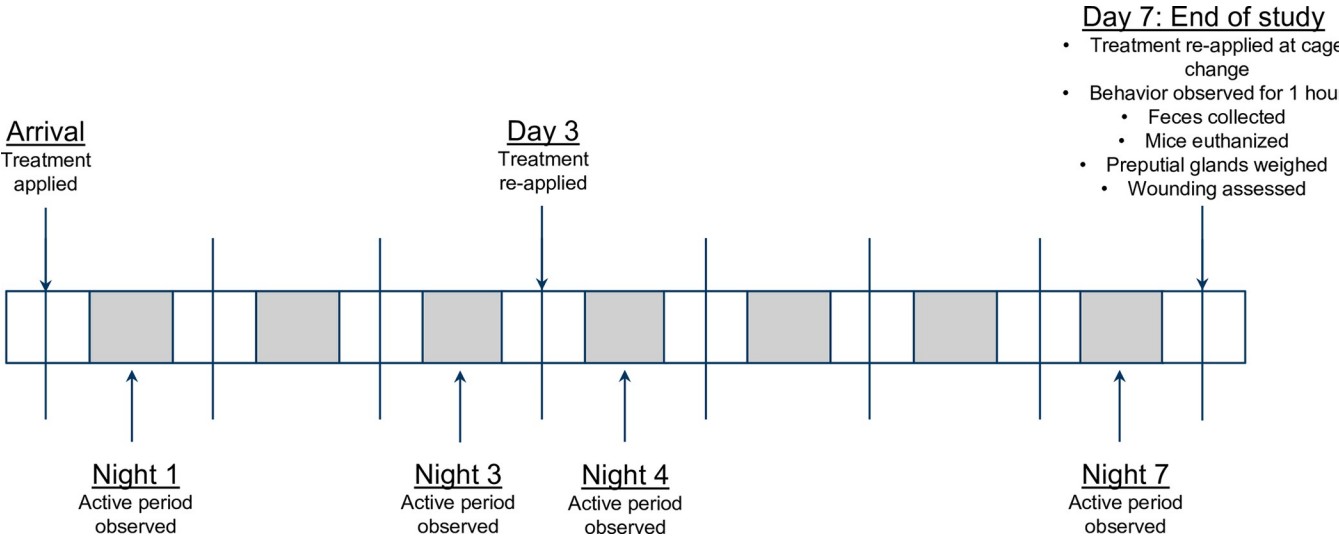

**Fig 1. Timeline of treatment administrations and study measures.** All study related procedures are listed under the appropriate day or night.

with the company, each spray pump distributes approximately 60μL of solution (2 sprays/ treatment). After treatment administration, cages were kept empty for at least ten minutes to allow the acetone to evaporate and nesting material to dry, leaving PEG bound to any test compounds on the nesting material.

Based on headspace levels of 6-hydroxy-6-methyl-3-heptanone (see above, "Treatment preparation"), treatments were refreshed on day 3 of the study. Each cage received an additional 120μL of their assigned treatment applied to 1g of fresh nesting material. The additional gram of treated nesting material again was allowed to dry, and acetone evaporate, for ten minutes in the housing room before being distributed to the mice. On study day 7, cages were cleaned with new cage bottoms, clean aspen, and 8g of fresh nesting material containing 120μL of the respective treatment. Like previous administrations, ten minutes passed between treatment application and transferring mice to the new cages. Fig 1 summarizes the timeline of all treatment administrations and measures.

## Home cage behavior

Mouse cages were placed on wire metro racks, in video booths made of white foam board (Office Depot, Boca Raton, FL) to reduce background movement as done previously [19]. Two shelves on each rack were used, and each shelf contained two cages, one of each strain. Video data were continuously recorded using infrared closed-circuit television cameras (HDview, Los Angeles, CA) and GeoVision surveillance software (Taipei, Taiwan). Social behavior was scored during the dark phase (18:00–6:00) using the following categories: escalated aggression, mediated aggression, submissive, and allo-grooming (Table 2). Data were collected using one-zero focal sampling every five minutes the first night after arrival (night 1), the night before the treatment refresher (night 3), the night after the treatment refresher (night 4), and the final night (night 7). Further, behavior was recorded for one hour after cage change (occurring approximately between 8:30–9:30 on day 7) as aggression can peak 15–45 minutes after cage change [9,10]. Two observers coded video (AJB and a trained undergraduate assistant). Cages were randomly assigned a numerical label to blind observers to treatment, and they were viewed in a random order. It was not possible to blind observers to strain due the differing coat colors between B6 and SJL mice. Ten 12-hour periods of video were used for training

**Table 2. Ethogram of behaviors observed during the study.** Definitions were taken from mousebehavior.org.

| Category | Behavior | Description |
|---|---|---|
| **Mediated Aggression** | Resource Theft | A mouse will approach another that is either eating a piece of food or chewing on a piece of bedding. The approaching mouse will then attempt to take the resource from the other's paws or mouth. It may or may not be successful. It is usually preceded by a social investigation and typically involves both mice tugging at the resource. |
| | Tail Rattling | The fast waving movements of the tail. This behavior may be partially obscured by bedding material, but can be detected by displacement of bedding near a mouse's tail. |
| | Thrust | The aggressor mouse will first threaten its target cage mate by thrusting its head and fore body towards its cage mate's head or body. The aggressor's paw may come in brief contact with the target, but otherwise no contact is made. |
| | Mounting | Attempts to mount another animal in the absence of intromission. Palpitations with forepaws and pelvic thrusts may be present. |
| | Chase | A mouse will chase a fleeing partner, but no biting occurs |
| Submissive | Submissive Upright | A posture where the animal will sit on its haunches in an upright position exposing the belly. The forepaws are off the ground and may stretch out its forepaws towards the threatening mouse. Mouse can also be laying on its side with one forepaw and one hind paw stretched toward the threatening mouse and its back touching the ground. |
| | Fleeing | This behavior is characterized by a mouse moving away from the mouse performing an aggressive behavior. Typically fleeing animals will run, but in a confined space may walk or turn first. Also score if the mouse turns away without locomoting. Only score if responding to an aggressive behavior (mediated/escalated). |
| **Escalated Aggression** | Bite | The aggressor mouse attacks the recipient with open mouth and appears to bite the recipient, or latches onto the recipient by his teeth, or forcefully touches the recipient who responds by jumping or fleeing quickly. Aggressor mouse may rush or leap at the victim. However, it also includes a mouse using its teeth to grab and tug on another's tail. Only score for the mouse that is biting, not the victims. |
| | Fighting | A behavior displayed by each animal when locked together. Separate behaviors are difficult to distinguish properly due to the fast rolling over and over seen with the animals kicking, biting, and wrestling. The initial victim retaliates towards the attacker and does not submit appropriately. Score for all mice actively involved in the fight. |
| **Allo-grooming** | | In this interaction, an actor mouse frequently uses its forepaws for stability when grooming the recipient. During grooming, the actor mouths and licks the fur on the recipient's body. The actor will also use its teeth to clean the hair shaft by pulling the fur from the base of the hair shaft upward or outward. |
| **Active** | | Score if the mouse is visible and moving for more than 5 seconds. |

representing 6.5% of the total video watched. Formal interrater reliability was calculated before coding began using Cronbach's alpha and was based on four observations periods (two per strain). Initial reliability scores are as follows: 0.97 (general activity), 0.93 (mediated aggression), 0.81 (escalated aggression), and 0.83 (allo-grooming). After coding was complete, reliability was assessed again using the last three observation periods viewed in the study. Final reliability scores are as follows: 0.97 (general activity), 0.81 (mediated aggression), 0.70 (escalated aggression), and 0.87 (allo-grooming). To replicate the methods used to identify the VOC and behavior correlations [19], all behaviors categorized as mediated aggression and submissive were initially coded as mediated aggression. However, in order to distinguish reactions to aggression from mediated behaviors, a single observer (AJB) recoded any instances of observed aggression to specify if submissive behaviors were performed. Hence, there is no reliability measure for submissive behaviors. From the video data, the proportion of active time in which each behavior category was observed was calculated per night per cage, as well as after cage change. These behavior measures are considered the primary outcome for this study.

## Fecal corticosterone metabolites

On day 7, fecal samples were collected by individually housing the mice in cages with a shallow layer of aspen bedding for two hours. Fecal corticosterone metabolites (FCM) increase approximately 8–10 hours after a spike in plasma corticosterone, if it occurs during a period when mice are mostly inactive [30]. Previous data from this lab has shown that aggression counts peak in the last two hours of the dark, active period (unpublished data). Sample collections

began between 13:30–14:00 to capture these final hours of the mice's active period, with most of the lag time occurring during the inactive period. Collecting during a limited time range also ensured that daily glucocorticoid fluctuations would not influence the data [31].

Afterwards, feces were gathered with metal forceps, placed in 1.5mL Eppendorf tubes, and stored in a -80˚C freezer until processing. Samples were only analyzed from each cage's dominant and subordinate mouse, as glucocorticoids are elevated in animals undergoing repeated social defeat [32–36]. Dominant and subordinate mice were determined by their preputial gland weight: body length ratio as this has been shown to align with individual conflict win/defeat patterns within a cage [37]. Glands were weighed in mg with an analytical balance (Ohaus, Parsippany, NJ) and body lengths were taken in mm with digital calipers. Since this measure is obtained after euthanasia, feces were collected from all mice, but only analyzed from the mice with the highest and lowest preputial gland ratio per cage. If any of those mice did not produce enough feces for analysis (at least 20mg dry weight), they were excluded. Across cages, 90% of dominant mice and 92% of subordinate mice produced enough feces for analysis, leaving N = 71 samples.

FCMs were analyzed using a previously described method [30]. Briefly, samples were dried at 80˚C for two hours, dry mass weights were obtained, and each sample was crushed to a powder. A 20–50 mg (depending upon availability) aliquot of each dry sample was weighted. Steroids were extracted by adding 1 mL of 80% methanol to the 50 mg of dry feces, or an aliquot in case of samples with less weight). Then samples were vortexed by hand for three 30 second periods and centrifuged for ten minutes at 2500 g. A portion (0.5 mL) of each methanolic supernatant was placed in a new Eppendorf tube and dried at 70˚C for two hours. Dried extracts were shipped to the University of Veterinary Medicine, Vienna (Vienna, Austria) for enzyme immunoassays. After redissolving them in 80% methanol and diluting (1:20) with assay buffer, an aliquot was analyzed (in duplicate) in a 5α-pregnane-3β,11β,21-triol-20-one enzyme immunoassay (details see: Touma et al., 2003), which has been successfully validated for use in mice [38].

## Wounding

After feces collection, mice were euthanized with prolonged $CO_2$ and carcasses were frozen. Wounding was assessed using the Pelt Aggression Lesion Scale (PALS; Gaskill et al., 2016). Briefly, pelts were gently separated from the carcasses and pinned to a dissection board at each limb. Photos of the subcutis were taken (Sony, Tokyo, Japan) and then evaluated using a 9 x 9 grid, which was overlaid on each pelt image. The grid was stretched from the base of the neck to the base of the tail. Each grid square was evaluated on a 0–4 scale in terms of percent of subcutis visible and wound severity. This scale has been previously described [39]: higher scores represent more visibility and severe damage. Each square was scored with the following equation:

PALS Grid Score = Severity Score x Visibility Score x 0.25.

The average anterior, middle, and posterior region scores were calculated using the three squares closest to the base of the neck, three in the center column of the grid, and three closest to the base of the tail, respectively. Posterior scores can distinguish aggression related wounding from ulcerative dermatitis [39], but this study served to validate these scores with behavior. For each mouse, PALS were averaged per region, then region averages summed across all the mice in the cage. This provided an overall level of wounding in each body region in a particular cage.

## Statistics

Missing data note: for behavior data, video from four cages on night one was excluded due to technical failure. These data points were balanced across strain, but were all from the same

treatment (3,4-dimethyl-1,2-cyclopentadione). Further, one mouse from a cage of SJL treated with the control solution was found dead the morning of treatment refreshment (day 3), so video was only analyzed from night 1 and 3. This mouse did not contain wounding that met the humane endpoint criteria, so the cause of death is likely unrelated to aggression. Escalated aggression levels in this cage from days 1 and 3 were between the 60–75 quantile of values observed in the study and the sum of posterior wounding in the cage was between the 50–55 quantile. Feces were not collected from this cage, but wound scores were included in the analysis. Ultimately, repeated measure behavior models contained N = 154/160 observations, cage change behavior models contain N = 39/40 observations, the wounding model contained N = 120 observation (3 pelt region sums x 40 cages), and the FCM model contained N = 71/80 observations.

All measures were analyzed with general linear mixed models. Strain, treatment, and the interaction were tested as fixed effects. Repeated measures behavior data also included study day as a fixed effect, as well as any 2-way interactions. The wounding model included pelt region and total proportion of escalated aggression performed in the cage as fixed effects and any 2-way interactions. The FCM model included dominance status and individual posterior PALS score along with any 2-way interactions. Any non-significant interactions were dropped from the final models due to a lack of orthogonal data. Facility was tested as a block and cageID nested in strain and treatment was tested as a random effect. Batch number served as a blocking factor and would typically be tested as a fixed effect. However, since the study was designed using incomplete blocks, the analyses would not run with batch as a fixed effect. It has been argued that blocking factors can be considered random if treatments are randomly assigned to incomplete blocks [40], which they were here. Any non-significant covariates or blocking factors were dropped from the final models. Model assumptions were evaluated post-hoc by examining the predicted by residual and normal Q-Q plots and transformations were made as needed. An exception was made for allo-grooming in the post cage change period. This behavior did not occur often during the observation period, so a Poisson regression was used to analyze behavior counts. Significant main effects were further analyzed with post hoc Tukey or student's t-tests. All analyses were done in JMP Pro (version 16.1.0). Significant P values from the behavior models were adjusted with the sequential Bonferroni correction to account for the multiple models assessing social behavior [41].

## Results

### Home cage behavior

**Active period- repeated measures.** Volatile treatment did not affect any active period behavior (see Table 3). All social behavior categories were significantly impacted by strain, while mediated aggression and allo-grooming were also impacted by study day (P <0.001). SJL mice performed more escalated, mediated, and submissive behavior than B6 mice (Fig 2A, 2B and 2D). However, B6 mice performed more allo-grooming than SJLs (Fig 2E). Mediated aggression and allo-grooming were performed less on study day 1 compared to days 3, 4, and 7 (Tukey: P<0.05, Fig 2C and 2F).

For models where treatment was not significant, the effect size and least significant number (LSN) needed for a significant outcome with 80% power are reported in Table 4.

**Cage change.** Escalated aggression, mediated aggression, and allo-grooming after cage change were not significantly altered by any factor in this study (Table 5). However, submissive behaviors were impacted by strain (Table 5), where SJL mice performed more than B6. Please refer to Table 4 for effect sizes and LSN calculations for the treatment predictor tested with mixed models. Since allo-grooming after cage change was analyzed with a Poisson regression,

**Table 3. Fixed effects and model $R_{adj}^2$ for each behavior measured across the study week (N = 154).**

| | Strain | Treatment | Strain*Treatment | Day | Model $R_{adj}^2$ |
|---|---|---|---|---|---|
| **Escalated aggression** | $F_{1,28.09} = 114.04$, $P_{adj} < 0.001$ | $F_{4,28.12} = 0.89$, $P = 0.484$ | $F_{4,28.06} = 1.36$, $P = 0.274$ | $F_{3,110} = 0.09$, $P = 0.967$ | 0.73 |
| **Mediated aggression** | $F_{1,27.41} = 48.89$, $P_{adj} < 0.001$ | $F_{4,27.42} = 0.65$, $P = 0.632$ | $F_{4,27.34} = 0.99$, $P = 0.429$ | $F_{3,109.8} = 7.65$, $P_{adj} < 0.001$ | 0.47 |
| **Submission** | $F_{1,29.28} = 212.21$, $P_{adj} < 0.001$ | $F_{4,29.31} = 0.77$, $P = 0.553$ | $F_{4,29.28} = 0.64$, $P = 0.636$ | $F_{3,110.8} = 0.87$, $P = 0.457$ | 0.92 |
| **Allo-grooming** | $F_{1,29.73} = 56.18$, $P_{adj} < 0.001$ | $F_{4,29.76} = 0.28$, $P = 0.887$ | $F_{4,29.73} = 0.51$, $P = 0.731$ | $F_{3,111.1} = 8.65$, $P_{adj} < 0.001$ | 0.84 |

Significant effects are shown in bold; $P_{adj}$ represents P values adjusted using the sequential Bonferroni correction

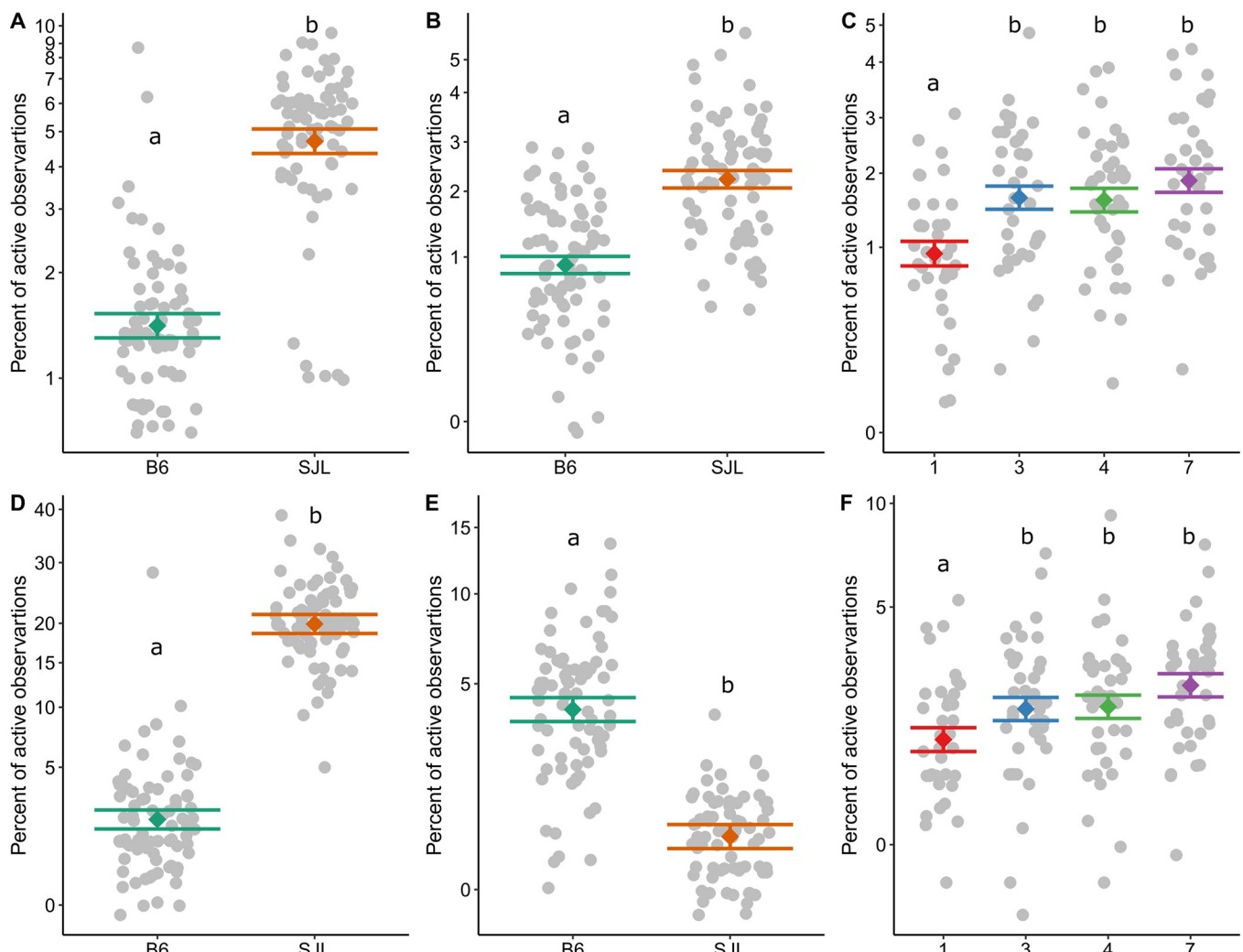

**Fig 2. Social behavior was affected by strain and study day.** SJL displayed more (A) escalated ($P_{adj} < 0.001$) and (B) mediated aggression ($P_{adj} < 0.001$). (C) Mediated aggression was also performed less on the first study day ($P_{adj} < 0.001$). (D) SJL mice performed more submissive behavior ($P_{adj} < 0.001$). (E) B6 mice performed more allo-grooming than SJL mice ($P_{adj} < 0.001$).). (F) Allo-grooming was also performed less on the first study day ($P_{adj} < 0.001$). All data are presented as factor level LSM ± SE with the scatter of individual residual error points (N = 154). Significant post hoc comparisons are indicated by differing letters within a panel. Y axes represent the percent of active time in which the behavior was observed. They are shown on a log10 back transformed scale in panel A, and a square root back transformed scale in panels B-F.

**Table 4. Effect size ($\eta M_p^2$) and least significant number (LSN) needed for a significant effect of treatment on each measure analyzed using mixed models.**

| | $\eta_p^2$ | LSN |
|---|---|---|
| **Escalated aggression- repeated** | 0.112 | 1041 |
| **Mediated aggression- repeated** | 0.087 | 656 |
| **Submission- repeated** | 0.095 | 1341 |
| **Allo-grooming- repeated** | 0.037 | 1098 |
| **Escalated aggression- cage change** | 0.104 | 173 |
| **Mediated aggression- cage change** | 0.045 | 386 |
| **Submission- cage change** | 0.129 | 261 |
| **Wounding** | 0.177 | 928 |
| **Fecal corticosterone metabolites** | 0.364 | — |

"—" indicates LSN not calculated as a significant effect was found.

the treatment effect size is reported here as the rate ratio for each factor level compared to the control: 3,4-dimethyl-1,2-cyclopentadione- 1.58; 3,5-diethyl-2-hydroxycyclopent-2-en-1-one- 0.95; 6-hydroxy-6-methyl-3-heptanone- 1.95; geranylacetone- 0.95.

## Wounding

Wounding was significantly altered by the interaction between the strain and pelt region ($F_{2,74} = 13.56$, P<0.001). The lowest wounding scores were seen in the anterior region of SJL cages (Tukey: P<0.05, Fig 3A). This was followed by scores in the anterior region of B6 cages and the middle region of SJL cages (Tukey: P<0.05, Fig 3A). The highest wounding scores were seen in the middle region in B6 cages and the posterior region of both strains (Tukey: P<0.05, Fig 3A). Wounding differences were also seen between pelt region and the proportion of time escalated aggression was observed while active ($F_{2,74} = 13.71$, P<0.001). Posterior wounding was higher as more escalated aggression was observed (Fig 3B; t(74) = 5.15, α/3, P<0.001). In contrast, anterior wounding was lower as more escalated aggression was observed (t(74) = -3.39, α/3, P = 0.001). The effect size and LSN for treatment are reported in Table 4.

## Fecal corticosterone metabolites

The concentration of FCMs was altered by strain ($F_{1,30.2} = 58.24$, P<0.001), treatment ($F_{4,25.81} = 3.69$, P = 0.017), posterior PALS score ($F_{1,49.87} = 8.14$, P = 0.006), and the treatment x average posterior PALS score interaction ($F_{4,46.48} = 4.69$, P = 0.003). B6 mice, regardless of treatment, had higher FCM than SJL mice (Fig 4A). For only mice treated with 3,4-dimethyl-1,2-cyclo-pentadione, FCM increased as posterior wounding increased (Fig 4B; t(54.98) = 3.68, α/5,

**Table 5. Fixed effects and model $R_{adj}^2$ for each behavior measured after cage change (N = 39).**

| | Strain | Treatment | Strain*Treatment | Model $R_{adj}^2$ |
|---|---|---|---|---|
| Escalated aggression | $F_{1,29} = 3.91$, P = 0.061 | $F_{4,29} = 0.83$, P = 0.512 | $F_{4,29} = 1.47$, P = 0.238 | 0.08 |
| Mediated aggression | $F_{1,29} = 2.32$, P = 0.139 | $F_{4,29} = 0.34$, P = 0.850 | $F_{4,29} = 2.16$, P = 0.098 | 0.09 |
| Submission | **$F_{1,29} = 31.07$, P<0.001** | $F_{4,29} = 1.08$, P = 0.386 | $F_{4,29} = 0.85$, P = 0.506 | 0.42 |
| Allo-grooming* | $\chi(1)< 0.01$, P = 0.976 | $\chi(4) = 3.29$, P = 0.511 | $\chi(4) = 2.24$, P = 0.692 | 0.10 |

Significant effects are shown in bold

"*" analyzed using Poisson regression, generalized $R^2$ is reported for the final model that contained only the main strain and treatment effects.

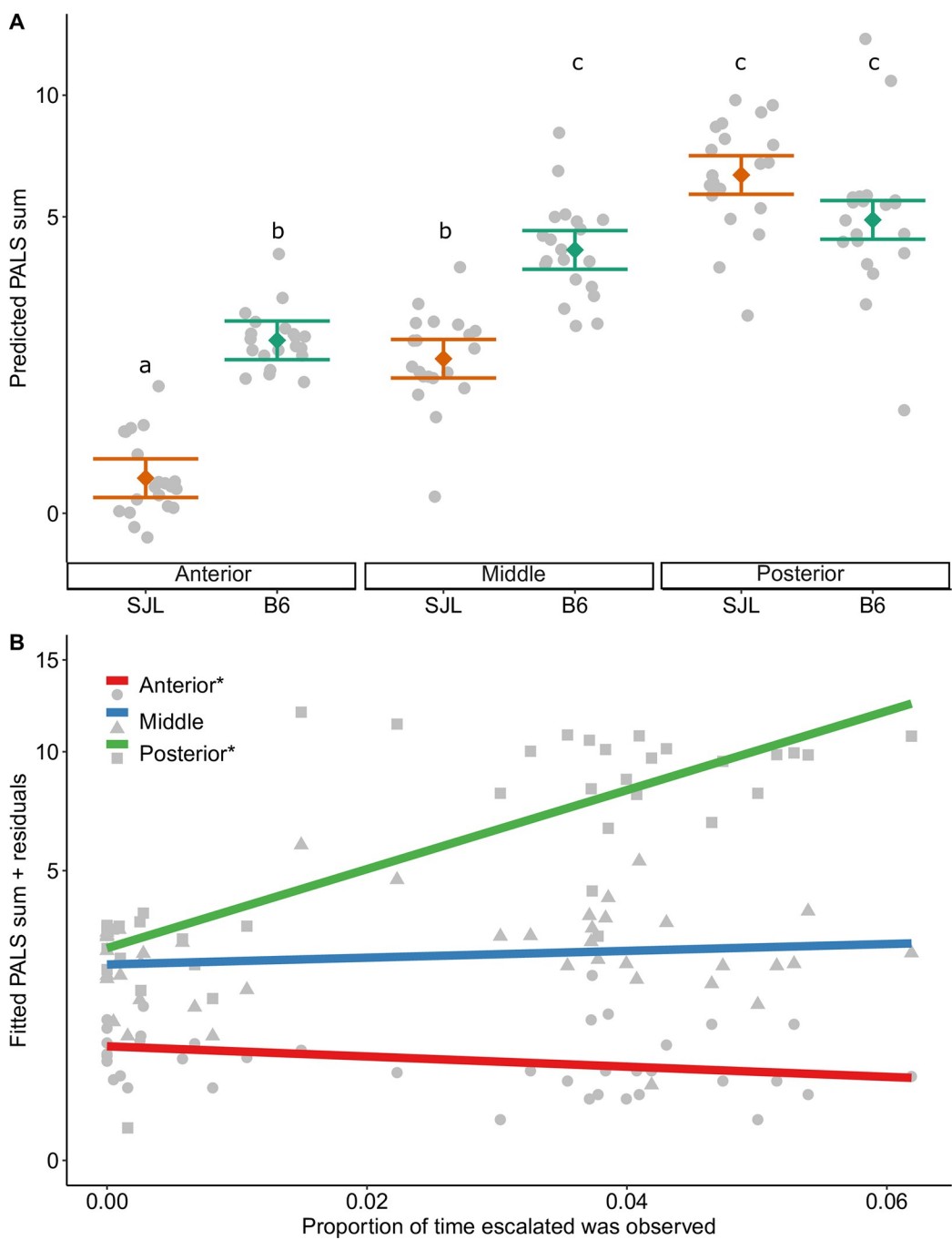

**Fig 3.** Wounding was impacted by (A) a strain x PALS region interaction and (B) a PALS region x proportion of escalated aggression interaction ($R_{adj}^2$ = 0.90, N = 120). Data are presented as factor level LSM ± SE with the scatter of individual residual error points in panel A. Significant post hoc comparisons are indicated by differing letters within each panel. In panel B, data are presented as the best fit line per PALS region over a scatter of individual residual error points. Slopes that significantly differ from zero are marked by an "*" in the legend. Y axes are shown on a square root back transformed scale.

P<0.001). However, mice that were treated with 3,5-diethyl-2-hydroxycyclopent-2-en-1-one, FCM decreased as wounding increased (t(40.10) = -2.82, α/5, P = 0.008). Overall, posterior wounding had a positive effect on FCMs (t(49.87) = 2.85, P = 0.006).

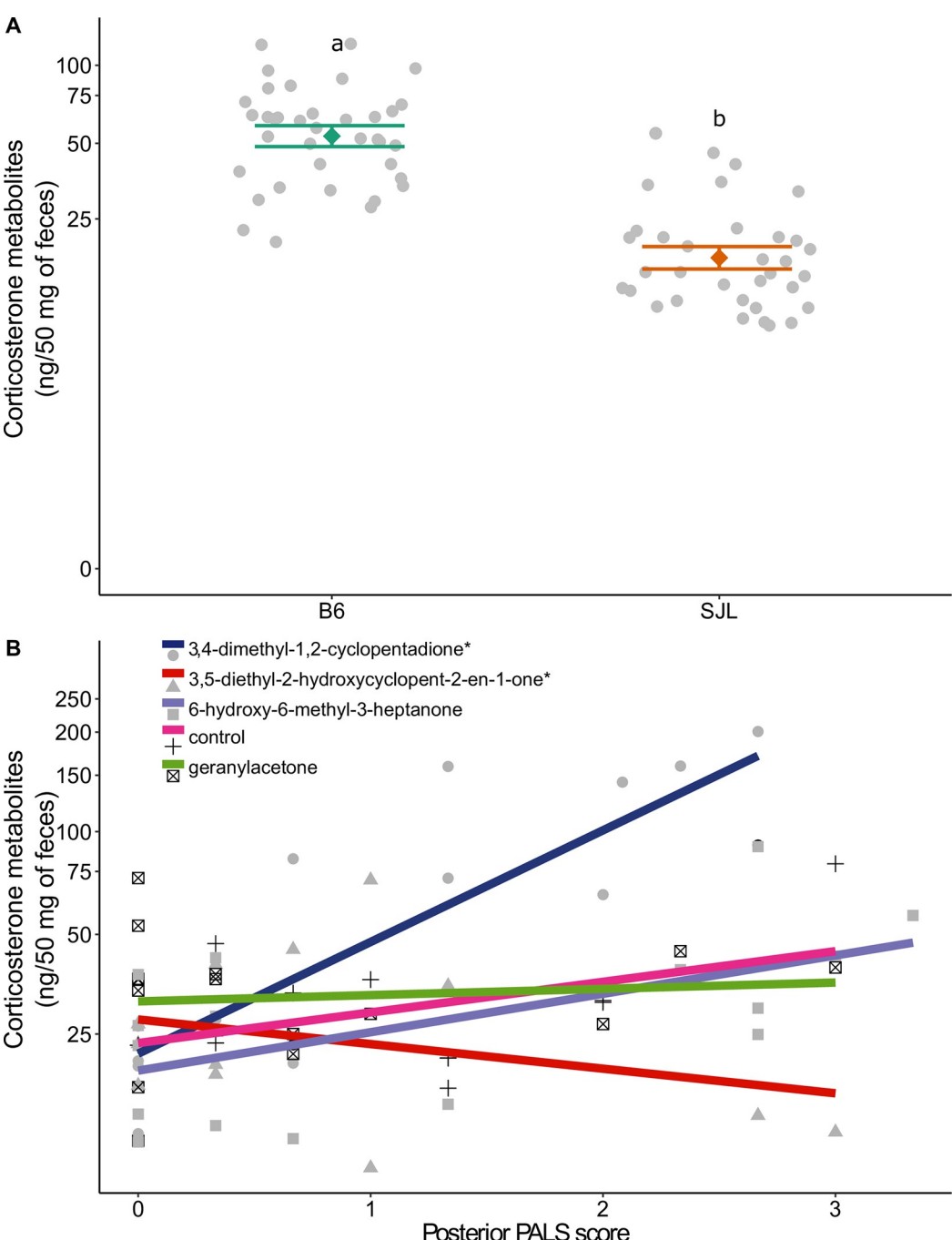

**Fig 4.** FCMs were impacted by (A) strain and (B) an interaction between posterior PALS score and treatment ($R_{adj}^2 = 0.66$, N = 71). Data are presented as factor level LSM ± SE with the scatter of individual residual error points in panel A. Significant post hoc comparisons are indicated by differing letters within a panel. In panel B, data are presented as the best fit line per treatment over a scatter of individual residual error points. Slopes that significantly differ from zero are marked by an "∗" in the legend. Y axes are shown on a log10 back transformed scale.

## Discussion

This study aimed to test whether VOCs that previously correlated with male mouse social behavior directly influence home cage interactions and if they could be considered murine

pheromones. Since geranylacetone negatively correlated with aggression [19], we expected it to reduce aggression here. We also expected 3,4-dimethyl-1,2-cyclopentanedione, 3,5-diethyl-2-hydroxycyclopent-2-en-1-one, and 6-hydroxy-6-methyl-3-heptanone to increase allo-grooming, since they previously correlated with this social behavior [19].

These data show that none of the VOC treatments tested here significantly altered social behavior in B6 or SJL mice. Based on $\eta_p^2$ calculations, these treatments had a small to intermediate statistical effect on most behaviors [42]. However, the LSN needed for a significant result is so large for each measure, that any biological effect is extremely weak and likely not worth investigating. This could be due to the confounding effect of strain on the previous correlations as both behavior and VOC levels were largely strain dependent [19]. Future endeavors could sample VOCs from cages with spontaneous occurrences of home cage aggression that are not so heavily strain biased. Further, the previous VOC datasets were reduced using Principal Component Analysis [19], and the components that explained the most variation were chosen to compare to behavior. It is possible that components with smaller explained variance had better predictive value [43] and their respective high loading VOCs should be further examined.

That being said, there were also several factors in this study that could have led to the null results found. In order to detect VOC levels in the headspace of the cage, 5x the natural concentration was used. Using such a high concentration not only rules out the possibility of confirming pheromone activity, but it could also have been high enough to alter a behavioral response [6,13]. Unfortunately, the true natural concentration of 3,4-dimethyl-1,2-cyclopentanedione and 3,5-diethyl-2-hydroxycyclopent-2-en-1-one have not been determined. Previously these compounds were identified in plantar sweat, which is produced in such low volumes that 1) pilocarpine is typically used to stimulate fluid production and 2) the samples were collected by directly rolling a Twister™ stir bar on the foot which did not permit fluid volume to be recorded [19]. While pilocarpine is often used in humans as a dry mouth remedy, there is individual variation in its effectiveness [44]. Further analytical work is needed on plantar sweat itself to determine how pilocarpine may impact VOC content, how much individual variation there is between mice injected with pilocarpine, and if VOCs can be collected without pilocarpine. This latter point would provide the most valid estimate of natural VOC concentrations in plantar sweat.

The application method could also have impacted the data seen here. The VOCs were administered in a 3% PEG, acetone solution as a first step to understand their efficacy at influencing behavior and to help rule out the effects of other molecules on behavior. However, two urinary murine pheromones known to increase inter-male aggression, 2-sec-butyl-thiazoline (SBT) and dehydro-exo-brevicomin (DHB), must be administered in castrate urine to provoke a behavioral response [14]. Both SBT and DHB are major urinary protein (MUP) ligands and need to interact with carrier proteins to be biologically active [45]. The same may be true for the VOCs tested here. It is possible that 3,4-dimethyl-1,2-cyclopentanedione and 3,5-diethyl-2-hydroxycyclopent-2-en-1-one must be administered in murine sweat to increase allo-grooming. However, collecting enough sweat for a treatment would be challenging as mice produce less than 100nL of sweat without pilocarpine stimulation [46] and creating a synthetic solution would not be possible without accurate compound concentrations. While the concentration of geranylacetone used here was based on the levels found in used nesting material, it originates in both murine sweat and saliva [19], so it may need another component from one of these fluids to be biologically active. Along those lines, 6-hydroxy-6-methyl-3-heptanone may need to be administered in castrate urine to increase allo-grooming; it is a known MUP ligand [27] and may need to interact with carrier proteins to be effective. It is currently unknown if the other three VOCs are protein ligands, but the possibility that they need a

transport protein cannot ruled out. Finally, SBT and DHB work synergistically to provoke a behavioral response [14]. It is possible that the VOCs tested here work in combination with one another, but this was not possible to test due to available time and resources.

While these specific compound treatments were not effective at improving male mouse social interactions, it cannot be denied that odor signals play a role in modulating home cage social behavior. General scent cue disruption can trigger aggression [8,47]. The most common example of this effect is routine cage cleaning, after which aggression peaks are often seen. However, preserving used nesting material at cage cleaning can reduce aggression peaks, and it has been shown that used nesting material contains a variety of protein associated odor signals used for identification purposes [11,12]. Since it is often recommended that male mice be kept in stable groups from an early age [1,48], perhaps odor profile familiarity is key for reducing aggression in the laboratory. Recognizing a cage mate's odor profile rather than individual appeasement odors may be sufficient to prevent fighting.

Social behavior was primarily impacted by strain, where SJL mice performed more aggression and submission while B6 mice performed more allo-grooming. These strain patterns are consistent with past work done by this group and another group's reported characterization of male SJL mice [19,37,49]. Interestingly, both mediated aggression and allo-grooming were performed less on the first study day than the others. This day effect was not previously reported, but past work found that cage level frequencies of allo-grooming are higher seven days after arrival compared to two days after (unpublished). The reduced levels of each behavior on the first night of the study may be because the mice were still acclimating to their new environment and spent less time engaging in these social behaviors. The similar pattern between these two behaviors is interesting as allo-grooming is often considered affiliative in mice [25]. Anecdotally, allo-grooming in this study was often followed by chasing as the recipient tried to end the grooming bout and the actor followed in pursuit. This aligns with past work showing a correlation between an individual's place in a grooming network and their place in a chasing, but not fighting, network [50]. This is not to suggest that allo-grooming is related to dominance, as the amount of allo-grooming performed and received did not predict social rank within the home cage [37].

Wound severity served as a secondary measure of escalated aggression and was impacted by an interaction between PALS region and the proportion of observed active time where escalated aggression was observed. At the cage level, wound severity in the posterior region increased with observed escalated aggression. This finding provides behavioral validation for past work showing that posterior PALS scores correctly predict fighting related wounding [39]. Further, wounding was impacted by an interaction between strain and PALS region. The highest scores were seen in the posterior region of both strains as well as the middle section of B6 mice. This may suggest that while most aggression is directed at the hindquarters, B6 mice may have a larger target area that extends into the middle of the back.

Despite the lack of treatment effects on behavior, there was a significant interaction between treatment and posterior wounding on FCMs. Rodents undergoing repeated social defeat are known to have elevated plasma corticosterone levels in both short- and long-term measurements [32–36]. The only treatment where this pattern extended to FCMs was 3,4-dimethyl-1,2-cyclopentanedione, despite similar wounding levels across treatments. It is unknown why this pattern was not seen in all mice, particularly the control mice. However, posterior wounding did have an overall positive effect on FCMs, implying that aggression related wounding has hormonal impacts that could alter a variety of research parameters. In contrast, mice treated with 3,5-diethyl-2-hydroxycyclopent-2-en-1-one had a negative relationship between wounding and FCMs. To the best of our knowledge, this pattern has not been documented before in mice. However, work in humans and non-human primates has

shown that hypocortisolism can be a consequence of chronic stress, potentially protecting individuals from the consequences of prolonged HPA axis activity [51,52]. It has been suggested that hypocortisolism in non-human primates can be an indicator of social stress [53], so a similar mechanism may explain these results in mice.

Finally, there was also a strain effect on FCMs: B6 mice had higher FCM concentration than SJL mice. Previous work has shown that strain can influence FCMs, with male B6 mice producing more FCMs than male BALB/c mice [54]. In female mice, the strain effect has been variable across studies using B6, BALB/c and DBA mice [55,56]. To the best of our knowledge, a comparison between male B6 and SJL mice has not been reported before.

## Conclusion

This study served as a follow up to previous work demonstrating a correlation between four VOCs and reduced aggression or increased affiliative behavior in group housed male laboratory mice. While the treatments in this study did not impact social behavior in the home cage, it is possible that the administration methodology could have altered the VOCs' biological activity. It is worth pursuing future work using concentrations closer to natural levels and in solvents that better represent the natural fluids in which these VOCs were detected. Further, it is possible that the tested VOCs were subjected to strain biases in the correlation study. Future sample analyses should focus on spontaneous occurrences of home cage aggression that are not so heavily strain biased.

## Supporting information

**S1 Table. Tested solution concentrations of the four compounds.**
(XLSX)

**S2 Table. Solution test order for each room.**
(XLSX)

**S1 Fig. Representation of how test solutions were applied in the empty cage.** The teal circle indicates where the test compound solution was placed, and the pink circles indicate where the stir bars were placed for collection.
(PDF)

**S1 Data. Raw data values from each cage for active period behavior, cage change behavior, fecal corticosterone metabolites, and wounding.**
(XLSX)

## Acknowledgments

We'd like to thank the following people for their contributions to this study: David Williams and Jacob Desmond of Indiana University for synthesizing 3,5-diethyl-2-hydroxycyclopent-2-en-1-one; Jonathan Karty, Robert Pepin, and Angela Hansen of the Indiana University Mass Spectrometry Facility for collecting and analyzing the cage headspace samples used for treatment preparation; Katie Bachert for help collecting fecal samples; and undergraduate assistant Stephanie Dijak for time spent coding video data.

## Author Contributions

**Conceptualization:** Amanda J. Barabas, Helena A. Soini, Milos V. Novotny, Jeffrey R. Lucas, Marisa A. Erasmus, Heng-Wei Cheng, Brianna N. Gaskill.

**Data curation:** Amanda J. Barabas, Rupert Palme.

**Formal analysis:** Amanda J. Barabas, Brianna N. Gaskill.

**Funding acquisition:** Helena A. Soini, Milos V. Novotny, Jeffrey R. Lucas, Marisa A. Erasmus, Heng-Wei Cheng, Brianna N. Gaskill.

**Investigation:** Amanda J. Barabas, Brianna N. Gaskill.

**Methodology:** Amanda J. Barabas, Helena A. Soini, Milos V. Novotny, Rupert Palme.

**Project administration:** Amanda J. Barabas.

**Supervision:** Jeffrey R. Lucas, Marisa A. Erasmus, Heng-Wei Cheng, Brianna N. Gaskill.

**Writing – original draft:** Amanda J. Barabas.

**Writing – review & editing:** Helena A. Soini, Milos V. Novotny, Jeffrey R. Lucas, Marisa A. Erasmus, Heng-Wei Cheng, Rupert Palme, Brianna N. Gaskill.

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
