## [Decision Letter · Decision Letter 0]

28 Jun 2022

PONE-D-22-13840Assessing the effect of compounds from plantar foot sweat, nesting material, and urine on social behavior in male mice, Mus musculusPLOS ONE

Dear Dr. Gaskill,

Thank you for submitting your manuscript to PLOS ONE. After careful consideration, we feel that it has merit but does not fully meet PLOS ONE’s publication criteria as it currently stands. Therefore, we invite you to submit a revised version of the manuscript that addresses the points raised during the review process.

We look forward to receiving your revised manuscript.

Kind regards,

Sankarganesh Devaraj

Academic Editor

PLOS ONE

Journal Requirements:

3. Thank you for stating the following financial disclosure: "This work was funded by a grant awarded to B.N.G, M.A.E., J.R.L., H.W.C., H.A.S., and M.V.N. by the Purdue University Center for Animal Welfare Science. Also, GC-MS data was analyzed using the Aligent GC-QTOF-MS purchased by NSF grant #1726633, awarded to Jonathan A. Karty. The funders had no role in study design, data collection and analysis, decision to publish, or preparation of the manuscript."

We note that one or more of the authors is affiliated with the funding organization, indicating the funder may have had some role in the design, data collection, analysis or preparation of your manuscript for publication; in other words, the funder played an indirect role through the participation of the co-authors. If the funding organization did not play a role in the study design, data collection and analysis, decision to publish, or preparation of the manuscript and only provided financial support in the form of authors' salaries and/or research materials, please do the following:

a. Review your statements relating to the author contributions, and ensure you have specifically and accurately indicated the role(s) that these authors had in your study. These amendments should be made in the online form.

b. Confirm in your cover letter that you agree with the following statement, and we will change the online submission form on your behalf: 

“The funder provided support in the form of salaries for authors [insert relevant initials], but did not have any additional role in the study design, data collection and analysis, decision to publish, or preparation of the manuscript. The specific roles of these authors are articulated in the ‘author contributions’ section.

6. Please include a copy of Tables 5.2 and E.1 which you refer to in your text on pages 6 and 11.

7. We note you have included a table to which you do not refer in the text of your manuscript. Please ensure that you refer to Table 2 in your text; if accepted, production will need this reference to link the reader to the Table.

Additional Editor Comments (if provided):

Dear Authors,

The reviewers have now commented on your manuscript. I suggest you to go through the comments and concerns raised by the reviewers. Most importantly, I personally found a lack of technical and scientific writing in the manuscript. Along with the reviewers comments, this could also be addressed/rectified.

Reviewers' comments:

Reviewer's Responses to Questions

**Comments to the Author**

1. Is the manuscript technically sound, and do the data support the conclusions?

Reviewer #1: No

Reviewer #2: Yes

2. Has the statistical analysis been performed appropriately and rigorously? 

Reviewer #1: I Don't Know

Reviewer #2: Yes

3. Have the authors made all data underlying the findings in their manuscript fully available?

Reviewer #1: No

Reviewer #2: Yes

4. Is the manuscript presented in an intelligible fashion and written in standard English?

Reviewer #1: No

Reviewer #2: Yes

5. Review Comments to the Author

Reviewer #1: In this manuscript, Barabas and Gaskill et al. described the effects of pheromonal compounds from plantar foot sweat, nest material, and urine on social behavior in male mice. The aim of the study is fascinating and essential for the welfare of experimental animals; however, the reviewer thought that the manuscript is suited to publishing journals related to experimental animals. The reviewer recommends making appropriate figures and tables for publication; moreover, the main text might be fully edited for the convenience of the readers.

Reviewer #2: Comments

The manuscript entitled “Assessing the effect of compounds from plantar foot sweat….. social behavior in male mice, Mus musculus” explores the VOCs correlate with behavior analysis. The flow of the manuscript is well written and provides useful information to the readers. The authors have carried out the follow-up work with well-structured data on the correlation of VOCs with mice behavior. The results are relevant with the objectives and presented well. The data are presented with appropriate photographs and tables. Further, the data were analyzed using suitable statistical approaches. However, the manuscript needs to address a few concerns which are listed below to improve the quality of manuscript.

Technical comments:

Abstract

4 previously identified compounds- include the compounds names in parentheses

Respective compound solution? Technically not right.

Home cage social behavior …. Of the animals?

Volatile treatments? Do you mean the compounds?

B6 or c57BL/6N- Words should be used uniform throughout the text

Introduction:

The rationale for choosing 6-hydroxyt-6-mnethyl-3-heptanone should be clearly mentioned. There have been many compounds reported to have various behavioral effects in conspecifics.

It is unclear from the methods that whether the authors used one compound/one time, or mixed all the compounds in one cage. This should be clearly mentioned in the methods. And, if not mixed, reasons should be provided. Because, in natural conditions, all those compounds are mixed in the bedding material. Therefore, keeping this in mind, discussion should be revised at appropriate instances.

Line no. 127-131. Appropriate figures could be included to enhance the readership

Line no. 167- what is CIS4?

Treatments- should be included as a flow chart! This will improve the understanding of the readers. It should include (type of treatment, day of treatment, end of treatment, etc.) in a chart. The authors could utilize ARRIVE guidelines flow chart for this purpose.

Line no. 212- sat for ten minutes? Rewrite the sentences, as these lines are non-technical. Similar non-technical words exist in the manuscript. I suggest the authors to correct it.

Table 2. include references (if applicable)

Fecal corticosterone metabolites

Is there any difference in the corticosterone conc. Between the first day and last day? Did the authors measure any differential data? Why it was done only on day 7?

Line no. 251-261- not necessary. But if the authors wish, they can trim these lines to a few and include.

If the preputial gland data is presented, it should come under a separate subtitle (not under fecal corticosterone). Originally, line no 274 is the start point of fecal corticosterone analysis.

Line no. 281- The details of commercial kits (lot no., supplier, sensitivity, specificity) should be included.

Line no. 342- P<0.001 not p values<0.001

Figure 1 legends should be appropriately marked, because all the figures shows percent of active observations.

Line no. 455- delete “discussed above”

Line no. 464- if this rationale was not tested, it is better to remove this inference.

Line no. 479- This fact was not reported in the results section.

Line no. 507- a variety of research parameters?

Line no. 510-514- discussion should be modified according to the presented results.

Line no. 515-519- the real reason for this difference should be discussed and further implications of these results should be included.

General comments:

Reframe the sentences in line 47 ‘Only recently has it been’ rephrase the lines since it is not clear.

In line 154, How the nesting material samples were set? Any rationale behind 0.58g?

Methods section contains various parameters like temperatures, degree symbols it is advised to recheck throughout the same. E.g., in line 163 the degree symbol is missing.

The authors are mentioned in line 189, the animals are sacrificed and weight were calculated. If so, how the animals were sacrificed?

The meaning of the abbreviations should be clearly defined at their mention in line no. 274.

Please ensure that your manuscript meets PLOS One Journal style requirements to reference. In particular, the article title and that contain the species name should be in italics in line numbers 561, 565, 586. And also check line 611, 618 and so on.

Based on the above comments I recommended major revision of the manuscript before accepted for publication.

The manuscript should be checked regarding the technical information. Authors need to improve the quality of the paper by improving both scientific and technical writing.

6. PLOS authors have the option to publish the peer review history of their article (what does this mean?). If published, this will include your full peer review and any attached files.

Reviewer #1: No

Reviewer #2: No

---

## [Author Response · Author response to Decision Letter 0]

11 Aug 2022

All reviewer comments have been addressed in an uploaded file.

---

## [Decision Letter · Decision Letter 1]

17 Oct 2022

Assessing the effect of compounds from plantar foot sweat, nesting material, and urine on social behavior in male mice, Mus musculus

PONE-D-22-13840R1

Dear Dr. Erasmus,

We’re pleased to inform you that your manuscript has been judged scientifically suitable for publication and will be formally accepted for publication once it meets all outstanding technical requirements.

Kind regards,

Sankarganesh Devaraj

Academic Editor

PLOS ONE

Additional Editor Comments (optional):

The comments and suggestions of the reviewer and Editor are considered, and the manuscript is improved. The manuscript, in its present form, accepted for publication.

Reviewers' comments:

Reviewer's Responses to Questions

**Comments to the Author**

1. If the authors have adequately addressed your comments raised in a previous round of review and you feel that this manuscript is now acceptable for publication, you may indicate that here to bypass the “Comments to the Author” section, enter your conflict of interest statement in the “Confidential to Editor” section, and submit your "Accept" recommendation.

Reviewer #2: All comments have been addressed

2. Is the manuscript technically sound, and do the data support the conclusions?

Reviewer #2: Yes

3. Has the statistical analysis been performed appropriately and rigorously? 

Reviewer #2: Yes

4. Have the authors made all data underlying the findings in their manuscript fully available?

Reviewer #2: Yes

5. Is the manuscript presented in an intelligible fashion and written in standard English?

Reviewer #2: Yes

6. Review Comments to the Author

Reviewer #2: The manuscript has been improved by incorporating appropriate changes based on the comments and suggestions. Besides, the language of the manuscript is also improved to the journal's standards. Taken together, I recommend accepting the manuscript in its present form.

7. PLOS authors have the option to publish the peer review history of their article (what does this mean?). If published, this will include your full peer review and any attached files.

Reviewer #2: No

---

## [Editor Report · Acceptance letter]

24 Oct 2022

PONE-D-22-13840R1 

Assessing the effect of compounds from plantar foot sweat, nesting material, and urine on social behavior in male mice, *Mus musculus*

Dear Dr. Erasmus:

I'm pleased to inform you that your manuscript has been deemed suitable for publication in PLOS ONE. Congratulations! Your manuscript is now with our production department. 

Kind regards, 

on behalf of

Dr. Sankarganesh Devaraj 

Academic Editor

PLOS ONE